# Ketone Bodies after Cardiac Arrest: A Narrative Review and the Rationale for Use

**DOI:** 10.3390/cells13090784

**Published:** 2024-05-04

**Authors:** Filippo Annoni, Elisa Gouvea Bogossian, Lorenzo Peluso, Fuhong Su, Anthony Moreau, Leda Nobile, Stefano Giuseppe Casu, Elda Diletta Sterchele, Lorenzo Calabro, Michele Salvagno, Mauro Oddo, Fabio Silvio Taccone

**Affiliations:** 1Department of Intensive Care, University Hospital of Brussels (HUB), 1070 Brussels, Belgium; 2Experimental Laboratory of Intensive Care, Department of Intensive Care, Free University of Brussels (ULB), 1070 Brussels, Belgium; 3Department of Anesthesiology and Intensive Care, Humanitas Gavazzeni Hospital, 24125 Bergamo, Italy; 4Department of Biomedical Sciences, Humanitas University, Pieve Emanuele, 20072 Milan, Italy; 5Medical Directorate for Research, Education and Innovation, Direction Médicale, Centre Hospitalier Universitaire Vaudois (CHUV), University of Lausanne, 1011 Lausanne, Switzerland

**Keywords:** ketone bodies, ischemia–reperfusion, cerebral metabolism, anoxic brain injury, heart arrest

## Abstract

Cardiac arrest survivors suffer the repercussions of anoxic brain injury, a critical factor influencing long-term prognosis. This injury is characterised by profound and enduring metabolic impairment. Ketone bodies, an alternative energetic resource in physiological states such as exercise, fasting, and extended starvation, are avidly taken up and used by the brain. Both the ketogenic diet and exogenous ketone supplementation have been associated with neuroprotective effects across a spectrum of conditions. These include refractory epilepsy, neurodegenerative disorders, cognitive impairment, focal cerebral ischemia, and traumatic brain injuries. Beyond this, ketone bodies possess a plethora of attributes that appear to be particularly favourable after cardiac arrest. These encompass anti-inflammatory effects, the attenuation of oxidative stress, the improvement of mitochondrial function, a glucose-sparing effect, and the enhancement of cardiac function. The aim of this manuscript is to appraise pertinent scientific literature on the topic through a narrative review. We aim to encapsulate the existing evidence and underscore the potential therapeutic value of ketone bodies in the context of cardiac arrest to provide a rationale for their use in forthcoming translational research efforts.

## 1. Introduction

The human brain, despite its modest size, is an avid consumer of energy. Remarkably, it utilises approximately 25% of the body’s available glucose and 20% of the systemic oxygen delivery [1,2]. Given its lack of significant energy storage capacity, the brain’s functionality hinges on a consistent and adequate supply of substrates, rendering it especially vulnerable to abrupt energy shortages.

Cardiac arrest (CA) represents a dramatic clinical condition where systemic perfusion is compromised, leading to a rapid depletion in the available energetic supply. Without the prompt restoration of organ perfusion, cells undergo profound metabolic failure, leading initially to deep cellular sufferance and culminating in cell death. Cardiopulmonary resuscitation manoeuvres reinstate a minimal organ perfusion in the attempt to delay this process while facilitating therapeutic interventions, such as electrical defibrillation and pharmacological support. Yet, even with the restoration of blood flow and cardiovascular function, a cascade of deleterious processes, including neuroinflammation, oxidative stress, and metabolic dysfunction, can exacerbate organ damage in the aftermath of CA [3]. Such a phenomenon is generally identified as the “post cardiac arrest syndrome” [4].

Despite its significant prevalence and the profound impact on mortality and morbidity, no specific treatment is available in the clinical practice to adequately reduce the burden of brain injury, and even therapeutic hypothermia, considered effective for a long time, has been recently challenged [5]. The ketogenic diet (KD) and ketone bodies (KBs) have gathered attention across a spectrum of medical conditions for their potential therapeutic benefits. KBs have been consistently linked with neuroprotective effects in preclinical research, with the mitigation of neuronal hyperexcitability, decreases in neuronal degeneration, and enhanced brain energetics, as recently reviewed [6]. Moreover, the administration of KBs has shown promising results in experimental models of, among others, traumatic brain injury [7] and acute cerebral ischemia [8].

While a vast body of the literature has explored their role in various acute pathologies, a paucity of research specifically addressing their utility in the context of CA remains.

In this manuscript, we provide an overview of KB metabolism, we resume the current knowledge on the subject by conducting a narrative review, and we discuss the potential favourable properties of KBs in the context of CA.

## 2. Metabolism of Ketone Bodies

The liver can synthesise up to 300 g of ketones daily [9]; this production primarily stems from the long-chain fatty acids stored in the adipose tissue. As illustrated in Figure 1, these fatty acids are mobilised as free fatty acids (FFAs) in response to reduced insulin release, a phenomenon observed during prolonged fasting or starvation [10]. FFAs are then oxidised to Acetyl-CoA, which can then feed into the tricarboxylic acid cycle to generate ATP. However, when the mitochondrial capacity to process Acetyl-CoA via the Krebs cycle is surpassed, ketone bodies (KBs)—specifically D/L-ß-hydroxybutyric acid and Acetoacetate—are produced from Acetyl-CoA. Given the liver’s inability to metabolise these compounds, they are released into the bloodstream, becoming accessible to various organs, including the brain and heart.

Besides the brain, KB synthesis have been described elsewhere, including astrocytes and brain cancer cells [11,12].

The brain’s ability to employ KBs as an energy substrate is well established [6,10]. Under typical conditions or brief fasting periods, the rate of KB production matches with their peripheral uptake. Consequently, plasma ketone levels do not normally exceed 0.3 mM [9]. High levels of circulating KBs, often referred to as hyperketonaemia, are characterised by circulating KB concentrations surpassing 0.5 mM [13].

The cerebral uptake of circulating KBs is dependent on the expression of monocarboxylic transporters (MCTs), which rapidly rise in response to hyperketonaemia [6]; as the KB levels rise, glucose availability to the brain diminishes [14]. In particular, glial and endothelial cells on the blood–brain barrier as well as astrocytes express the isoforms with relatively low affinity for BHB (MCT1 and MCT4), while neurons mostly possess the MCT2 isoform [15,16]. Expressions of these transporters can be upregulated in rodents through both fasting and exercises [17,18]. Of note, previous preclinical research has described the enhancement of MCT expression following transient global ischemia [19], as well as after permanent unilateral mean cerebral artery occlusion [20], and, in a model of focal ischemia 48 h after unilateral extradural compression, MCT 1 and MCT 2 expression increased in both the ipsilateral and contralateral areas of the brain [21].

Intriguingly, the brain’s KB uptake is proportional to their circulating concentration, irrespective of the glucose levels [22], and, for such a reason, it is often considered that they constitute a preferred energetic substrate for the brain.

However, elevated circulating KB levels are also notoriously associated with pathological conditions. Diabetic ketoacidosis, a condition arising from diminished insulin levels relative to circulating glucose, is a case in point; typically observed in type 1 diabetic patients who cease insulin administration, it results in a paradoxical hyperglycaemic state in which glucose remains inaccessible to cells [23]. Consequently, KB production escalates, often exceeding 10 mM, associated with metabolic acidosis, hyperglycaemia, and dehydration due to osmotic diuresis [23]. Furthermore, during ketoacidosis, a decline in CoA transferase (Succinyl-CoA:3-oxoacid-CoA transferase, or SCOT) activity, which is vital for ketolysis, is observed in many cells, including cardiomyocytes. Since SCOT-dependent extra-hepatic oxidation represents a key pathway for the metabolisation of liver-derived KBs, the net effect represents the reduced peripheral capability of KB usage. This is attributed to the overexpression of glucose transporters, which downregulates ketone oxidation [6,24]. The accumulation of poorly metabolised β-hydroxybutyric acid (ßOHB) and acetoacetic acid in the bloodstream results in bicarbonate depletion and a subsequent pH drop, creating a potentially lethal medical emergency.

Contrastingly, nutritional ketosis, a physiological response to fasting or starvation, is not coupled with pH alterations, and is typically associated with a modest elevation in circulating KBs over a relative long period of time (days). Even when induced by exogenous KB supplementation, concentrations rarely surpass 5 mM in healthy adults [25]. Beyond the ketogenic diet, which is deemed unpleasant and poses challenges for maintenance in the long term, KBs can be effectively supplemented either orally or intravenously. This supplementation occurs in the presence of normal insulin and glucose levels, unlike the physiological conditions during fasting, starvation, or exercise.

As mentioned before, the potential therapeutic benefits of exogenous KB supplementation have gathered significant attention, especially in the context of various pathologies, including acute illnesses [26].

## 3. Cerebral Metabolism Is Impaired following Cardiac Arrest

In the proximity of the return of spontaneous circulation (ROSC) after CA, commonly with other critical conditions, glucose regulation disturbances are often observed, culminating in hyperacute hyperglycaemia [27,28]. Several endogenous as well as exogenous factors might influence the capacity to metabolise glucose in humans, such as increased cortisol production, catecholamines, and glucagon [27], as well as the administration of epinephrine during resuscitation manoeuvres. Those events favour lipolysis and ketogenesis, but liver function is often impaired after CA, although the clinical impacts of such phenomena is not well described [29,30].

Following ROSC, the post-resuscitation phase is marked by profound and enduring metabolic disturbances, including metabolic acidosis, hyperlactatemia, electrolyte imbalances, and impairments in glucose homeostasis [31], as pointed out by the evidence provided from experimental cardiac arrest (CA) studies, which have revealed that a significant number of animals exhibit a secondary rise in the cerebral lactate/pyruvate (L/P) ratio, often reported as a marker of the redox state of cells, suggesting severe energetic failure due to ischemia and mitochondrial dysfunction [32]. Moreover, this metabolic disruption is only partially ameliorated by therapeutic hypothermia [32], and the cellular metabolic shift towards lactate accumulation and the rise in the L/P ratio can be the result of increased aerobic glycolysis, a situation that occurs when, despite oxygen availability, the cell continues to produce ATP through glycolysis, with the minimal or absent involvement of the tricarboxylic acid cycle, or can be due to mitochondrial impairment [33]. To this regard, in a murine investigation, cerebral positron emission tomography scans (PET-CT) conducted 72 h after CA demonstrated an elevated uptake of 2-deoxy-2-[^18^F]fluoro-D-glucose (^18^F-FDG) in mice when compared to control groups [34], suggesting an increased metabolic demand in the proximity of a global cerebral anoxic injury. Furthermore, compromised mitochondrial function has been observed both during cardiopulmonary resuscitation (CPR) and following experimental CA, despite optimal resuscitation efforts [35,36].

Post resuscitation, elevated blood glucose levels and their fluctuations have been associated with an unfavourable prognosis [37]. However, the experimental evidence regarding this association remains a topic of debate [38,39]. Notably, experimentally induced variations in pre-arrest glucose levels have been linked to altered gene expression in resuscitated animals after cardiac arrest, suggesting that hyperglycaemia could induce a different early cerebral response that could be implicated in pathological processes [40]. Additionally, in acute brain injury patients, hyperglycaemia has been associated with an increased risk of death and poor neurological outcomes [41]. Intriguingly, a rodent study examining prolonged CA found that, while metabolic disturbances in the heart and liver were rectified after 30 min of cardiopulmonary bypass resuscitation, these disturbances intensified in the brain and kidneys [42]. This suggests a differential organ response to reperfusion following CA. A fundamental role in cellular metabolism is attributed to mitochondria, which produce ATP through oxidative phosphorylation, and thereby assure the continuity of cellular physiologic functions. Extensive preclinical and translational research has established a connection between ischemia–reperfusion events and mitochondrial dysfunction. Interestingly, alterations in mitochondria phospholipids have been deemed specific to the brain and not peripheral mitochondria, suggesting an increased vulnerability to ischemia–reperfusion [43], and, after CA, damaged mitochondria could release cytochrome C and apoptotic signals, thus promoting necrosis [44]. Moreover, excessive mitochondrial elimination through selective autophagy (mitophagy) has been described in an experimental model of asphyxial CA in rats [45].

In summary, both during CPR and throughout the post-resuscitation phase, the brain endures severe metabolic alterations. These alterations may contribute to the progression of cerebral injuries, even when perfusion is adequately restored.

## 4. Rationale in Cardiac Arrest

The human brain, while resilient in many ways, is particularly vulnerable to hypoxic-ischemic injuries. Recent research has highlighted the potential therapeutic benefits of KBs in addressing these challenges. The following paragraphs provide a comprehensive look at the multifaceted benefits of KBs in this context.

In normal conditions, KBs do not present a more favourable metabolic profile than glucose, having a less favourable phosphate to oxygen ratio (2.50 vs. 2.58, representing the amount of ATP produced per oxygen atom reduced by the respiratory chain) [46,47]. Nevertheless, insulin resistance is present after CA [27,31], and it is known that insulin plays a key role in the activation of pyruvate dehydrogenase in multiple tissue [48,49,50] which leads to pyruvate dehydrogenase inactivation, decreasing the overall cellular capacity to convert pyruvate to acetyl-CoA to fuel the tricarboxylic acid cycle. [46]. in such conditions, KBs represent a direct source of acetyl-CoA, and thus could represent a readily usable energetic substrate.

### 4.1. Glucose Sparing Effect

KBs provide an alternative energy substrate for the brain, allowing available glucose to be redirected from the glycolytic pathway to the pentose phosphate pathway, which is crucial for cellular membrane regeneration [51]. By inhibiting hepatic gluconeogenesis, KBs reduce glucose production and lipolysis, while maintaining stable insulin and glucagon levels [52]. This results in a favourable cardiovascular profile. In fact, in a randomised controlled trial in humans, the increase in ßOHB induced by lipopolysaccharide infusion was associated with a reduced cerebral glucose uptake and an increased cerebral blood flow, without a tangible effect on cerebral oxygen uptake [53]. Figure 2 illustrates the glucose-sparing effect in the brain and in the liver, induced by the presence of ketone bodies.

### 4.2. Seizure Control, Modulation of Oxidative Stress and Inflammation

Both the ketogenic diet and medium-chain triglyceride (MCT) supplementation have been effective in managing resistant epilepsy in children [54,55]. Post-cardiac arrest (CA) patients often experience epileptic activity [56], which is linked to unfavourable outcomes [57,58]. KB supplementation has shown anti-seizure activity [59,60], suggesting its potential as a coadjutant therapeutic strategy for epilepsy [61].

KBs, especially ßOHB, play a pivotal role in reducing oxidative stress via influencing histone acetylation [6,62] and blocking the inflammasome-mediated inflammatory response [63]. ßOHB also decrease reactive oxygen species production following glutamate excitotoxicity [64], and similar results have been reported in an experimental model of traumatic brain injury [65] and during hypoglycaemic conditions [66].

In recent years, KBs have been linked with both pro- and anti-inflammatory properties as follows: acetoacetic acid (AcAc) has been associated with increased endothelial injury [67], and ketosis has been associated with type 1 diabetes with increased inflammation [68], whereas ßOHB seems to exert anti-inflammatory properties in a variety of tissues via hydrocarboxylic acid receptors (HCA) [69]. In a culture of murine microglial cells, ßOHB suppresses LPS-induced inflammation by modulating NF-kß [70] and was associated with the recruitment of a neuroprotective population of monocytes/macrophages in a rodent model of a stroke [71]. Furthermore, ketone bodies have been deemed to mediate inflammation in both experimental models of heart failure and cardiovascular disease [72,73].

### 4.3. Sodium Load and Osmolarity

Ketone bodies, derived primarily from medium-chain fatty acids (MCFA)—abundantly found in breast milk—have gathered significant attention in the medical community. Medium-chain triglycerides (MCT) serve as a noteworthy alternative to the ketogenic diet, especially in paediatric patients with refractory epilepsy [74]. Beyond dietary sources, ketone bodies can be introduced directly into the system, either orally or intravenously in the form of esters or salts. While ketone salts are bound to minerals like sodium, esters are linked to alcohols [75]. Although ketone esters are typically favoured to prevent an excessive sodium load, the implications of a sodium-rich solution become particularly significant in the aftermath of CA.

A pivotal concern post-cerebral reperfusion is neuronal swelling. This phenomenon, believed to be triggered by the activation of voltage-gated chloride channels, results in a rapid influx of negatively charged ions into the cellular environment. This ion imbalance prompts a sharp increase in intracellular sodium, leading to a substantial water influx, causing cellular swelling and distress [76]. Notably, both hypo- and hypernatremia, observed upon arrival following out-of-hospital CAs, have been associate with worst neurological outcomes [77]. However, there is a silver lining: hypertonic saline combined with starches has shown promise, with a higher rate of returning spontaneous circulation in a human trial [78]. This suggests that a controlled sodium load for a limited period of time, when paired with ketone body salt administration, might mitigate the cellular membrane sodium gradient, potentially curbing neuronal swelling during the acute phase.

Furthermore, injectable salt esters of ßOHB can be formulated into hyperosmotic solutions. In the context of post-CA syndrome, such solutions might offer relief from episodes of elevated intracranial pressure, as previously documented [79,80]. Additionally, they may play a role in enhancing cerebral blood flow [81]. Supporting this notion, a study involving an electrically induced CA model in pigs found that hypertonic crystalloids were associated with improved cerebral perfusion pressure, increased mean arterial pressure, and reduced intracranial pressure when compared to isotonic counterparts [82]. Moreover, hypertonic–hyperoncotic solutions have been linked to decreased cardiac and astroglial injury markers post CA in pigs [83].

### 4.4. Improvement of Cardiac Function

In the immediate aftermath of CA, the prognosis remains grim. Within the initial three days following a CA, approximately one-third of patients who are admitted alive to the hospital die due to cardiovascular complications [84]. This high mortality rate is unsurprising given that a significant proportion of CA is precipitated by underlying cardiac pathologies, and that the recurrence of these conditions is most probable in the proximity of the acute event. Moreover, a substantial number of CA survivors must deal with cardiac impairment during the post-resuscitation phase [4], which can compromise both cerebral and systemic perfusion, thus triggering a cascade of detrimental events that often culminate in adverse clinical outcomes.

Emerging experimental evidence underscores the potential cardioprotective properties of KBs. For instance, KBs have demonstrated protective effects in models of ischemia–reperfusion injury [85,86], and have shown promising results in a porcine model of myocardial infarction [87]. Furthermore, a recent clinical trial revealed that the infusion of ßOHB resulted in notable cardiac and hemodynamic improvements without adversely affecting cardiac mechano-energetics in both chronic heart failure patients and their healthy counterparts [88]. Collectively, the data suggest that interventions such as dietary-induced ketosis, ßOHB infusion, or ketone ester supplementation, could improve recovery in ischemic hearts [89].

Taken together, the body of evidence advocates for the therapeutic potential of exogenous KB supplementation. Such interventions could offer a robust energy source during metabolic dysfunction, supporting cellular repair processes. Whether they can be considered for acute neuroprotection rather than neurological recovery enhancement remains an interesting investigational challenge that is yet to be clarified.

## 5. Ketone Bodies and Cardiac Arrest

Recent advancements in the field of cardiac arrest research have shed light on the potential therapeutic benefits of empagliflozin, a novel inhibitor of the sodium-glucose co-transporter 2 (SGLT2) commonly prescribed for diabetes management. In a rat model of cardiac arrest, Tan et al. explored the effects of empagliflozin, drawing some intriguing observations. Rats treated with this drug exhibited enhanced myocardial function, prolonged survival, and a notable reduction in myocardial fibrosis, troponin-I levels, and oxidative stress when compared to their untreated counterparts. Notably, empagliflozin administration was associated with elevated myocardial ketone concentrations and ßOHB-lysis-associated myocardial gene expression [90].

In another experimental setup, rats subjected to transient cerebral ischemia induced by the occlusion of all four cerebral vessels demonstrated intriguing outcomes when exposed to a 48 h fasting regimen. These animals exhibited a significant reduction in neuronal death across various brain regions 72 h post ischemia, accompanied by decreased brain lactate concentrations [91]. Furthermore, KB supplementation was also proven useful in reducing cerebral damage in a similar model of total cerebral ischemia as follows: rats that received an isotonic ßOHB infusion either 30 min prior to ischemia induction (at a dose of 50 mg/Kg/h) or immediately post induction (at 30 mg/Kg/h) exhibited extended survival, diminished cerebral water and sodium content, and increased cerebral ATP levels [92].

While these transient global ischemia models offer valuable insights into global anoxic brain injuries similar to those seen in cardiac arrest, it is crucial to note the potential differences. The systemic ischemia–reperfusion events characteristic of CA, coupled with profound cardiovascular dysfunction, might attenuate, or even neutralise, the observed benefits of ketone body supplementation seen in vessel occlusion-based cerebral ischemia models. Hence, it is imperative to conduct studies that accurately replicate the multifaceted physiological disturbances associated with CA resuscitation.

In an effort to consolidate the existing literature on the potential therapeutic role of KBs in CA, a literature search was undertaken. Searches were conducted according to the PRISMA guidelines for systematic reviews [93] across Medline, Embase, and CENTRAL databases as of 01 October 2023. The search sought to specifically identify studies in which ketone bodies or ketogenic diets have been tested as interventions in the context of cardiac arrest, and the search was not intended to collect data on a specific outcome to perform a secondary analysis. The results were not restricted for language, and only articles published on scientific journals were included in the review, aiming to gather a comprehensive overview. Using MeSH terms, keywords, and Boolean operators “AND” and “OR”, the aim was to pinpoint studies where ketone bodies or the ketogenic diet were evaluated as interventions specifically in the context of cardiac arrest. Language restrictions were not imposed, and only articles from scientific journals were considered. The detailed search strategy for each database is available in Appendix A.

Two independent investigators (FA and EGB) screened the identified manuscripts. Discrepancies were resolved through discussions. From the initial 88 articles, 11 duplicates were removed. Based on title and abstract reviews, 68 articles were excluded. Of the remaining nine articles assessed in-depth, four were excluded, resulting in five articles that met the inclusion criteria for this review, as illustrated in Figure 3.

The included studies, detailed in Table 1, were experimental designs involving small animals, predominantly rodents. Notably, three studies emanated from the same research group [94,95,96]. All articles were published after 2007. In all but one study, the effects of a ketogenic diet (spanning 25 or 28 days) were compared against a standard diet [94,95,96,97]. One study employed the intraperitoneal injection of KBs (i.e., ßOHB) prior to inducing cardiac arrest [98]. Typically, the sample sizes were small, with most studies involving 4–6 animals [94,95,96,98]. An exception was one study with seventeen animals subjected to a ketogenic diet, followed by CA [97]. All studies were planned to investigate the neuroprotective properties of KBs in the proximity of a CA of various origins (chemical, mechanical, or hypoxic), lasting between 8 min and 8 min and 30 s.

The outcome measures varied across studies, encompassing the development of post-resuscitation seizures, the susceptibility to myoclonic jerks [94], the number of Fluoro-Jade-stained neurons as an expression of neuronal necrosis in different cerebral regions [95,96], survival rate, behavioural tests, and neurological scores [97,98], as well as the degree of apoptosis and mitochondrial fission [98]. Notably, only one study administered KBs following the return of spontaneous circulation, which could limit the clinical applicability of these findings [98].

In summary, irrespective of the administration method or dietary regimen, all studies indicated a beneficial effect of mild ketosis on neurological outcomes, suggesting a potential neuroprotective role of ketone bodies against brain injury and dysfunction following CA. However, the limited number of studies, small sample sizes, and inherent limitations of animal models underscore the need for further research before translating these findings into clinical practice.

## 6. Conclusions

Supplementing with KBs has shown promise in providing robust neuroprotective benefits, ideally tailored for the body’s requirements during the acute and the recovery phase post-resuscitation following CA. However, the existing literature is sparse, primarily consisting of small-scale preclinical studies conducted on small animals. Despite the limited scope, these findings collectively inspire further exploration into clinically relevant and translational research.

## Figures and Tables

**Figure 1 cells-13-00784-f001:**
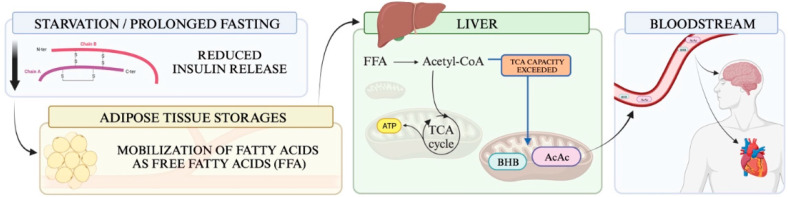
The metabolic pathway of ketone body production during states of reduced insulin release. A decrease in insulin triggers the release of fatty acids in the form of free fatty acids (FFAs). These FFAs are then converted into Acetyl-CoA through oxidation. When the capacity of mitochondria to utilise Acetyl-CoA in the tricarboxylic acid (TCA) cycle is exceeded, a different metabolic route is engaged, leading to the creation of ketone bodies. Once formed, these ketone bodies are dispersed into the bloodstream and delivered to various organs, including the brain and heart, for energy use. FFAs: free fatty acids; BHB: D/L-ß-hydroxybutyric acid; AcAc: acetoacetic acid; TCA: tricarboxylic acid cycle.

**Figure 2 cells-13-00784-f002:**
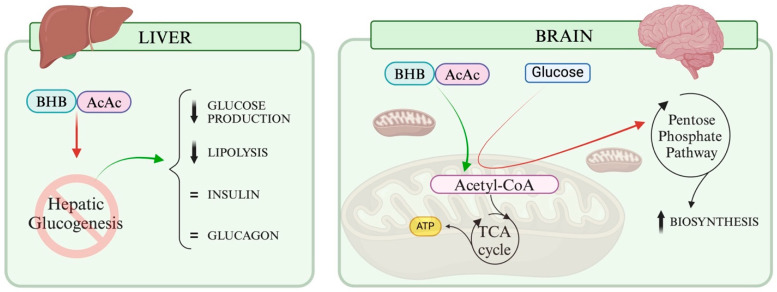
Glucose-Sparing Effect in the brain and in the liver by Ketone Bodies. Ketone bodies (KBs) serve as an alternative energy source for the brain and the liver, thereby conserving glucose, which can be redirected to the pentose phosphate pathway allowing an increase in biosynthesis. In the liver, KBs inhibit hepatic gluconeogenesis, reducing glucose production and lipolysis, while maintaining stable insulin and glucagon levels.

**Figure 3 cells-13-00784-f003:**
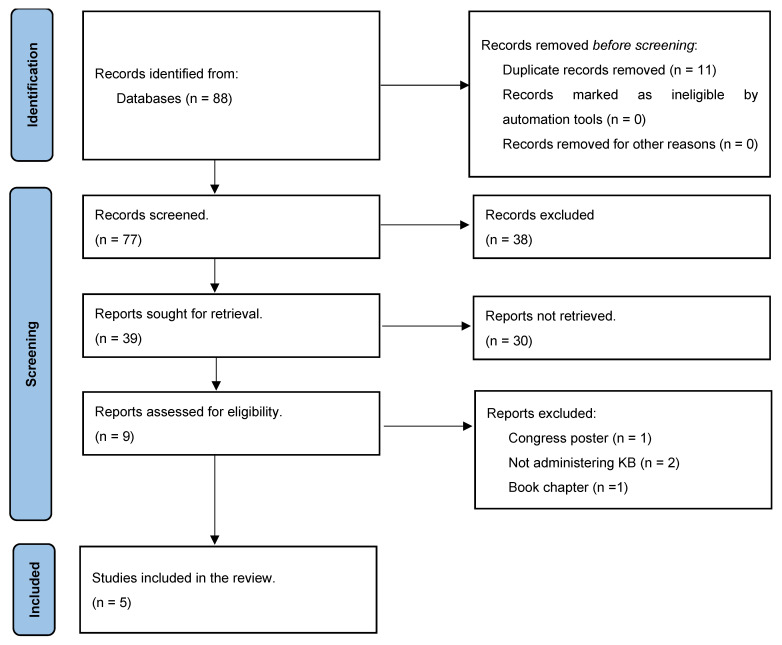
Flowchart of the narrative review.

**Table 1 cells-13-00784-t001:** Studies including KB administration in cardiac arrest.

Author	Year	Title	Species/CA	Intervention	Groups	Main Findings
Tai KK [94]	2007	Ketogenic diet prevents seizure and reduces myoclonic jerks in rats with cardiac arrest-induced cerebral hypoxia	Rat, mechanical (8′)	Ketogenic diet (KD, 25 days) vs. Standard diet (SD)	1—Ketogenic diet (n = 5) 2—Standard diet (n = 5)	Rats on SD developed seizures within 24 h and none in the KD. Animals in the SD group developed more severe myoclonic jerks induced by auditory stimuli when compared to the KD group.
Tai KK [95]	2008	Ketogenic diet prevents cardiac arrest-induced cerebral ischemic neurodegeneration	Rat, mechanical (8′30″)	Ketogenic diet (KD, 25 days) vs. Standard diet (SD) +/− levetiracetam	1—SD (sham, n = 4) 2—SD/CA (Control, n = 4) 3—KD/CA (n = 4) 4—SD-Levetiracetam/CA (negative Control, n = 4)	KD decreased the number of Fluoro-Jade-stained neurons in the hippocampus, cerebellum, and thalamus
Tai KK [96]	2009	Intracisternal administration of glibenclamide or 5-hydroxydecanoate does not reverse the neuroprotective effect of ketogenic diet against ischemic brain injury-induced neurodegeneration	Rat, mechanical (8′30″)	Standard diet (SD) or Ketogenic diet (25 days) +/− intracisternal injection of glibenclamide or 5-hydroxydecanoate 4 h before CA	1—SD/no CA (n = 6) 2—SD/CA (n = 6) 3—KD/CA (n = 6) 4—KD/Glibenclamide (n = 6) 5—KD/5-hydroxydecanoate (n = 6) 6— SD/amiloride (n = 6) 7— SD/saline (n = 6)	Fluoro-Jade-stained neurons in the hippocampus and cerebellum were lower in the KD exposed rats, regardless the intracisternal injection of glibenclamide or 5-hydroxybenzoate, confirming the neuroprotective effect of KD and suggesting that KATP channels do not play a significant role in KD-mediated neuroprotection
Peng F [97]	2022	Ketogenic diet attenuates post-cardiac arrest brain injury via the upregulation of pentose phosphate pathway-mediated antioxidant defence in a mouse model of cardiac arrest	Mouse, chemical (K+) (8′)	Ketogenic diet (KD, 4 weeks) vs. Standard diet (SD)	1—KD/CPR (n = 17) 2—SD/CPR (n = 14) 3—KD/Sham (n = 6) 4—SD/Sham (n = 6)	KD-CA mice had improved survival, better neurological score and behavioural tests compared to SD/CPR. KBs reduce glucose utilisation in the brain, suppress ROS production and activate pentose phosphate pathway.
Tan Y [98]	2022	Ketone body improves neurological outcomes after cardiac arrest by inhibiting mitochondrial fission in rats	Rat, asphyxia, (8′)	ßOHB (200 mg/Kg) intraperitoneal injection or saline	1—Sham (n = 6) 2—Control (n = 6) 3—ßOHB (n = 6)	ßOHB-treated animals showed higher survival (72 h), lower lactate levels (6 h), improved neurological function (72 h), reduced disarrangements of neurons in the CA-1 hippocampal area, and increased neuron numbers compared to the Control and Sham groups. Moreover, ßOHB decreased apoptosis, pyroptosis in neurons, and reduced mitochondrial fission, while improving mitochondrial function.

KD = ketogenic diet; SD = standard diet; CA = cardiac arrest; CPR= cardiopulmonary resuscitation; KATP = ATP-sensitive potassium channels; ßOHB = beta-hydroxybutyrate.

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
