# Peer review of "Ketone Bodies after Cardiac Arrest: A Narrative Review and the Rationale for Use"

_cells, 2024, doi:10.3390/cells13090784_

Round 1
Reviewer 1 Report
Comments and Suggestions for Authors
Summary:
This review article presents a review of the potential therapeutic effects of ketone bodies after cardiac arrest. The main focus appears to focus on cerebral function and brain injury.
Comments, Concerns, and Suggestions:
1. There are paragraphs immediately beneath Figure 1 (page 2) and Figure 2 (page 5) that appear to be relatively similar to the previous paragraph. Are these paragraphs errors or are then supposed to be included as part of the Figure Legend. If part of the Figure Legend, this would make sense and actually be appropriate since the Figure Legends are rather uninformative. There is no legend provided for Table 1.
2. The focus of the article is somewhat confusing, particularly when considering the title. While ‘cardiac arrest” is certainly a systemic issue, the majority of the references and discussion in the manuscript seems to focus more on the cerebral consequences rather than the heart. Perhaps a small revision in the title would provide more clarity to the reader.
a. Related to this point, the major section which seems to be the main focus of the article “Ketone Bodies and Cardiac Arrest” represents a small portion of the total article. The number of references discussed is low and the relevance of some of the articles (especially on page 7) is questionable.
3. Some additional details regarding the search criteria is required or the information regarding the methodology should be removed (as well as Figure 3). The inclusion and exclusion criteria is not clear and there is no specific details provided in the figure legend.
a. Why were 38 records excluded?
b. What does “reports not retrieved” mean? Why were they not retrieved.
c. In the 2nd to last paragraph on page 7, the authors briefly discuss reference #92 as study that demonstrated positive effects on KB supplementation on cerebral damage. Why was this study not included in the “systematic review” results? Seems like the systematic review has some flaws.
4. In the second paragraph of Section 5 (page 5), the authors suggest that insulin resistance results in pyruvate dehydrogenase dysfunction, but do not provide a direct reference. The reference included is a review article. This statement requires further explanation.
a. Also, the P/O ratio for glucose is generally reported as 2.58 (see Karwi, et al, 2018)
5. In Section 5.1, paragraph 1, the authors indicate the “KB reduce glucose production and lipolysis” which “results in a favorable cardiovascular profile”. How? No reference supporting this claim is provided.
6. The last paragraph in Section 5.5 (page 7) provides some potential effects of KB on a variety of effects but no references are provided.
7. The first four “paragraphs” on page 4 are only 1-2 sentences each. Seems like the content is similar among them so condensing these “paragraphs” into one paragraph is necessary.
8. There are some sections that are overly brief and could be deleted from the manuscript.
a. Section 4 (page 4-5) is very brief and does not seem critical to the overall discussion. This information could be condensed into a sentence or two as part of the introduction.
b. Section 5.2 (page 6) is also brief, and the relevance is not clear. Again, a brief sentence acknowledging that ketone bodies have been used as a therapy for epilepsy in the introduction would be sufficient.
9. There are some editorial issues that need to be addressed:
a. Title of Section 2: “Ketone Body Metabolism” or “Metabolism of Ketone Bodies”
b. Page 3, line 6 – The abbreviation for “BBB” was not defined.
c. Section 5, page 5, paragraph 2 – the last sentence of this paragraph is incomplete.
d. Figure 2 (liver panel) – “Lypolisis” should be “Lipolysis”
e. Section 5.3, page 6, paragraph 2 – Is there a word missing or is this a spacing issue?
f. Throughout the manuscript there is an occasional use of contractions, which is generally not used in scientific writing.
Comments on the Quality of English LanguageThe quality of the English language is fine, however, there are several editorial errors that need to be addressed.
Author Response
This review article presents a review of the potential therapeutic effects of ketone bodies after cardiac arrest. The main focus appears to focus on cerebral function and brain injury.
We would like to thank the reviewer for the timely and attentive response to our manuscript. Overall, we have used your remarks to improve the quality of the manuscript, and we believe it is now stronger.
Comments, Concerns, and Suggestions:
- There are paragraphs immediately beneath Figure 1 (page 2) and Figure 2 (page 5) that appear to be relatively similar to the previous paragraph. Are these paragraphs errors or are then supposed to be included as part of the Figure Legend. If part of the Figure Legend, this would make sense and actually be appropriate since the Figure Legends are rather uninformative. There is no legend provided for Table 1.
We would like to thank the Reviewer for this observation. The two paragraphs are indeed the figure legends. For the table 1, a legend is added to Tab 1.
- The focus of the article is somewhat confusing, particularly when considering the title. While ‘cardiac arrest” is certainly a systemic issue, the majority of the references and discussion in the manuscript seems to focus more on the cerebral consequences rather than the heart. Perhaps a small revision in the title would provide more clarity to the reader.
We would like to thank the Reviewer for this comment. While Cardiac arrest is most often caused by a cardiac pathology, only a minority of resuscitated patients die in the following days due to cardiac problems. As the matter of facts, up to 2/3 of patients will eventually die due to cerebral complications, and those are also the main determinants of long-term morbidity. That is why new treatment that seek to minimize the impact of CA shall aim at first in saving the brain rather than the heart (for which effective causative treatment exists). On the other hand, we agree that the title is rather vague in its current form, and we modified so that it refers to the period after cardiac arrest, so that the accent is now in the post-resuscitation phase.
- Related to this point, the major section which seems to be the main focus of the article “Ketone Bodies and Cardiac Arrest” represents a small portion of the total article. The number of references discussed is low and the relevance of some of the articles (especially on page 7) is questionable.
We would like to thank the Reviewer for this comment. It is indeed unfortunate that an attentive reader cannot see that CA is indeed the main focus of our work. The chapter 3 discuss the general metabolic alterations in CA, the chapter 5 the rationale behind KB administration in CA, with a figure, and the chapter 6 the actual experimental evidence existing, with a resuming table (6 out of 9 pages). We agree that the existing literature is extremely poor and characterized by very preliminary works, but nevertheless the scope of our review is to identify the current evidence, even if questionable so far.
- Some additional details regarding the search criteria is required or the information regarding the methodology should be removed (as well as Figure 3). The inclusion and exclusion criteria is not clear and there is no specific details provided in the figure legend.
We would like to thank the Reviewer for this comment. The string of search used is provided in the supplementary material for all databases. We have indeed used a systematic approach, as suggested by the PRISMA guidelines, but we did not include systematic in the title. The review could nevertheless qualify as a scoping review, but we still decided to limit the title of the manuscript to a "narrative review" since a deeper search of the grey literature in this context wasn't conducted and seemed redundant for the scope of the manuscript. Some more explanations about inclusion and exclusions criteria were missing, and we have add them to the text.
- Why were 38 records excluded?
- What does “reports not retrieved” mean? Why were they not retrieved.
We would like to thank the Reviewer for this comment. We referred to the following manuscript: BMJ 2021; 372 doi: https://doi.org/10.1136/bmj.n71, which reports the PRISMA 2020 updated guideline for reporting systematic reviews. We have indeed used the term suggested in the guidelines. It is not recommended to provide explanations on why the records are excluded, because generally the number is too high, and the search restitute records (abstracts or titles) in which some terms might be present but where exclusions criteria are obvious (pediatric or qualifying as posters, or reviews etc). Concerning the point "b" the same document define report as:" Report—A document (paper or electronic) supplying information about a particular study. It could be a journal article, preprint, conference abstract, study register entry, clinical study report, dissertation, unpublished manuscript, government report, or any other document providing relevant information".
Normally it refers to the complete article, rather than the title or abstract (Records). Information that are not present in the abstract might be present in the methods section or anywhere else, but is not recommended to provide explanations for the first passages.
- In the 2ndto last paragraph on page 7, the authors briefly discuss reference #92 as study that demonstrated positive effects on KB supplementation on cerebral damage. Why was this study not included in the “systematic review” results? Seems like the systematic review has some flaws.
We would like to thank the Reviewer for this comment. Reference 92 is not included in the "systematic review", because the animals did not experience cardiac arrest, but transient cerebral ischemia, so did not qualify within our search terms. It is nevertheless discussed in our manuscript, since the two mechanisms of brain damage are similar, yet not identical (no whole-body ischemia-reperfusion, or reduced cardiac function etc). Since we opted for a narrative review, we have removed the "systematic" word from the text.
- In the second paragraph of Section 5 (page 5), the authors suggest that insulin resistance results in pyruvate dehydrogenase dysfunction, but do not provide a direct reference. The reference included is a review article. This statement requires further explanation.
We would like to thank the reviewer for this comment. We added a sentence to better clarify the statement and added the reference.
- Also, the P/O ratio for glucose is generally reported as 2.58 (see Karwi, et al, 2018)
We would like to thank the reviewer for this comment. We agree that the sentence was incorrect, and the numbers were inverted. We have also added the reference.
- In Section 5.1, paragraph 1, the authors indicate the “KB reduce glucose production and lipolysis” which “results in a favorable cardiovascular profile”. How? No reference supporting this claim is provided.
We would like to thank the reviewer for this observation. Nevertheless, the text referred is the legend of the figure, not the text that refers to it. This was due to the editing process and will be hopefully clearer in this new version. The refences are reported in the text and are the 50-51 and 52.
- The last paragraph in Section 5.5 (page 7) provides some potential effects of KB on a variety of effects but no references are provided.
We would like to thank the reviewer for this observation. All the effects were discussed in the sections, so we have just listed them there, but given the redundance of those lines, we have removed them.
- The first four “paragraphs” on page 4 are only 1-2 sentences each. Seems like the content is similar among them so condensing these “paragraphs” into one paragraph is necessary.
We would like to thank the reviewer for this comment. We have reformulated the paragraph
- There are some sections that are overly brief and could be deleted from the manuscript.
- Section 4 (page 4-5) is very brief and does not seem critical to the overall discussion. This information could be condensed into a sentence or two as part of the introduction.
We would like to thank the reviewer for this comment. We have eliminated the section and add two sentences in the introduction.
- Section 5.2 (page 6) is also brief, and the relevance is not clear. Again, a brief sentence acknowledging that ketone bodies have been used as a therapy for epilepsy in the introduction would be sufficient.
We would like to thank the reviewer for this comment. We agree that the section is brief, but it has major clinical implications, and we have decided to unite with the following paragraph.
- There are some editorial issues that need to be addressed:
- Title of Section 2: “Ketone Body Metabolism” or “Metabolism of Ketone Bodies”
We would like to thank the reviewer for this observation. We have corrected it.
- Page 3, line 6 – The abbreviation for “BBB” was not defined.
We would like to thank the reviewer for this observation. We have corrected it.
- Section 5, page 5, paragraph 2 – the last sentence of this paragraph is incomplete.
We would like to thank the reviewer for this observation. We have corrected it.
- Figure 2 (liver panel) – “Lypolisis” should be “Lipolysis”
We would like to thank the reviewer for this observation. We have corrected it.
- Section 5.3, page 6, paragraph 2 – Is there a word missing or is this a spacing issue?
We would like to thank the reviewer for this observation. It was a spacing issue; we have corrected it.
- Throughout the manuscript there is an occasional use of contractions, which is generally not used in scientific writing.
We would like to thank the Reviewer for this observation. We have removed all contractions found in the text.
Reviewer 2 Report
Comments and Suggestions for Authors
To the authors,
The article “Ketone Bodies in Cardiac Arrest: A Narrative Review and The Rationale for Use” is a clear and well-structured manuscript. However, I have some questions and suggestions about it.
1. The conclusions of the manuscript are a bit vague and do not apport anything new to the topic. It is well-known the neuroprotective function of the ketone bodies in several brain issues. Moreover, you do not deep enough or compare, if it is possible, the main results of the five manuscript you include in the revision. Please, apport something new or innovative to the scientific society.
2. In the section two “Ketone Bodies’ metabolism” you talk about the KB metabolism in physiological and disease situations, such as diabetes. I would be clearer to add a reference to the pathological situations or split in two sections. Moreover, you focus on the mitochondrial synthesis of KB, however the complete synthesis pathway has been described for KB (Arnedo et al 2012; Montgomery C et al 2012) with a special focus in the brain tissue. If you are going to talk about the KB metabolism you should included all the metabolic pathways they are involved. Please add it.
3. The paragraphs after the two figures’ legends are a repeated idea of the previous paragraphs. I am not sure if it is a mistake or if they belong to the figure legend. Please clarify it.
4. Please remove the abbreviations written in italics of the end of page 2 (“FFA: Free Fatty acids; BHB: D/L-ß-Hydroxybutyric acid; AcAc: Acetoacetic acid; TCA: Tricarboxylic Acid Cycle “).
5. The are some abbreviatures which have not been defined previously, for example: BBB (in page 3), 18F-FDG (page 4) or TBI (page 6). Please recheck the whole manuscript.
6. Some of the references are old, for example references 7 and 8. Please, update them if it is possible.
Best regards,
Author Response
To the authors,
The article “Ketone Bodies in Cardiac Arrest: A Narrative Review and The Rationale for Use” is a clear and well-structured manuscript. However, I have some questions and suggestions about it.
We would like to thank the reviewer for the timely and attentive response to our manuscript. Overall, we have used your remarks to improve the quality of the manuscript, and we believe it is now stronger.
- The conclusions of the manuscript are a bit vague and do not apport anything new to the topic. It is well-known the neuroprotective function of the ketone bodies in several brain issues. Moreover, you do not deep enough or compare, if it is possible, the main results of the five manuscript you include in the revision. Please, apport something new or innovative to the scientific society.
We would like to thank the reviewer for this comment. We have reformulated the conclusions of the manuscript, hopefully to be more encouraging, to balance between the consistently different feedbacks received. On the other hand, the scope of a narrative review is to reorganize the existing literature around a subject, which appears to be extremely scarce. The current available experimental studies are hard to compare between them (different duration of CA, different techniques, different KD strategy), and three of them comes from the same author.
To our knowledge, there is no other manuscript describing the interesting properties of Ketone Bodies in the setting of CA, but our "apport to the scientific society" will be nevertheless limited by the scant and poor literature on the subject. In other words, we cannot conclude on something that is not supported in the literature. On the other hand, we believe that other researchers could use our review to plan new experimental designs in the field of cardiac arrest research.
- In the section two “Ketone Bodies’ metabolism” you talk about the KB metabolism in physiological and disease situations, such as diabetes. I would be clearer to add a reference to the pathological situations or split in two sections. Moreover, you focus on the mitochondrial synthesis of KB, however the complete synthesis pathway has been described for KB (Arnedo et al 2012; Montgomery C et al 2012) with a special focus in the brain tissue. If you are going to talk about the KB metabolism you should included all the metabolic pathways they are involved. Please add it.
We would like to thank the Reviewer for this observation. The reference number 20 (now 24) resumes the features of the diabetic ketoacidosis for the clinician. We have reported it before in the text. Regarding the synthesis of the KB, we believe that the message for the reader shall cover the most important pathways, rather than all existing pathways (such as peroxisomes in cancer cells such as in Mongomery 2012). Nevertheless, it is true that we have focused almost exclusively on the liver synthesis and we add a sentence and a reference upon the cerebral (astrocyte) synthesis of KB.
- The paragraphs after the two figures’ legends are a repeated idea of the previous paragraphs. I am not sure if it is a mistake or if they belong to the figure legend. Please clarify it.
We would like to thank the Reviewer for this observation. It is an editing problem; they belong to the figure legends.
- Please remove the abbreviations written in italics of the end of page 2 (“FFA: Free Fatty acids; BHB: D/L-ß-Hydroxybutyric acid; AcAc: Acetoacetic acid; TCA: Tricarboxylic Acid Cycle “).
We would like to thank the Reviewer for this observation. We have corrected it, this sentence also belongs to the figure legend.
- The are some abbreviatures which have not been defined previously, for example: BBB (in page 3), 18F-FDG (page 4) or TBI (page 6). Please recheck the whole manuscript.
We would like to thank the Reviewer for this observation. We have corrected it.
- Some of the references are old, for example references 7 and 8. Please, update them if it is possible.
We would like to thank the Reviewer for this observation. We have corrected it.
Reviewer 3 Report
Comments and Suggestions for Authors
I found this review to be highly informative and relevant; even though my field is in the area of TBI and stroke. The review is quite technical, but appropriate for readers who are versed in basic and translational research. It might be more difficult for some clinicians to absorb all the specific details and molecular biology; but from what I could discern, it was written with researchers in mind. This review has the potential to generate more interest in the use of ketones as therapeutic agents. I think that the authors did not 'overreach' in their conclusions.In fact, they were quite cautious. I would think that this paper could generate more interest in testing ketones in other models of cardiac injury, as well as for other disorders of the brain such as TBI or stroke.
Comments on the Quality of English Language
The quality of English in this paper is good, but very technical; so it might not attract attention from professions involved mainly in clinical practice. However, this is not a flaw from a scientific perspective. Readers may not always agree with the author's perspective, but they can learn a lot and develop new hypotheses as to how ketones may be neuroprotective in other forms of CNS or PNS injury.
Note: I did NOT check for plagiarism as this is the direct responsibility of the journal editors. The references do seem to be in order and relevant to the papers. The tables and figures (especially some of the tables) werre interesting and added to the perspective of the authors. I would recommend publication.
Author Response
We would like to thank the Reviewer for the kind and encouraging words.
Round 2
Reviewer 1 Report
Comments and Suggestions for Authors
Comments, Concerns, and Suggestions:
1. In the initial review, there was a concern regarding the title of the article and whether it accurately reflects the content of the article. The authors acknowledged the vagueness of the title and responded by changing the previous word “in” to “after”. This extremely minor change in the title does not adequately address the initial concern.
2. In the initial review, the section of “Ketone Bodies and Cardiac Arrest”, which appears to be the primary focus of the article, was questioned due to the relatively small portion of the total article and the relatively few references. The authors acknowledged that the “existing literature is extremely poor and characterized by very preliminary works”, thus confirming the reviewer’s point. Are 5 relevant published articles sufficient for an entire review article, particularly nearly half of the section discusses the “review criteria” for which little detail is provided (see below for additional concerns)? The authors should expand this area and provide deeper discussion of the existing literature and better highlight the information provided in Table 1. Importantly, the authors make no mention that the existing literature is sparse.
3. In the initial review, there were questions and concerns regarding the “literature search”, including the information presented in Figure 3. The authors’ reply has not addressed or alleviated this concern. The relevancy and need for the inclusion of the details of the search as well as Figure 3 or Supplementary Table is not clear. Approximately 3 paragraphs, which is nearly half of the seemingly main point of the review article, is dedicated to description of the “methods” for which the literature search was completed. In Figure 3, the authors show that nearly 70 potential records were excluded or not retrieved but provide no indication as to why these records were not included in the review. Did any of these articles show that ketone bodies were not beneficial in recovery from cardiac arrest? The figures, table, and section of the manuscript that discusses this “search” does not seem necessary without further explanation or rationale.
